# Evaluation of the Effect of the ENSO Cycle on the Distribution Potential of the Genus *Anastrepha* of Horticultural Importance in the Neotropics and Panama

**DOI:** 10.3390/insects14080714

**Published:** 2023-08-18

**Authors:** Arturo Batista Degracia, Julián Ávila Jiménez, Anovel Barba Alvarado, Randy Atencio Valdespino, Mariano Altamiranda-Saavedra

**Affiliations:** 1Instituto de Innovación Agropecuaria de Panamá (IDIAP), C. Carlos Lara 157, Ciudad del Saber 0843-03081, Panama; 2Facultad de Ciencias, Universidad Pedagógica y Tecnológica de Colombia, Avenida Central del Norte 39-115, Tunja 150003, Colombia; julianleonardo.avilajimenez@gmail.com; 3Institute of Agricultural Innovation of Panama/National Research System of Senacyt-Panama, Panama City 0816-02852, Panama; anovelbarba@gmail.com (A.B.A.); randy.atencio@gmail.com (R.A.V.); 4Grupo de Investigación Bioforense, Tecnológico de Antioquia Institución Universitaria, Medellín 050005, Colombia; maltamiranda2@gmail.com

**Keywords:** geographic distribution, ENSO *Anastrepha*, horticulture, neotropics, Panama

## Abstract

**Simple Summary:**

The genus *Anastrepha* (Diptera: Tephritidae) is a group of fruit flies that contain significant insects (pests) of agricultural importance since they alter the quality and reduce the yield of fruit and vegetable production. This in turn projects great uncertainty regarding safety due to quarantine restrictions around the world. In this work, potential distribution models are presented for four selected species of this genus of fruit flies in different scenarios of climate variability; these models can constitute an input for the design of preventive control plans and early warning systems for these pest insects. The models were designed using the MaxEnt Maximum Entropy algorithm and showed a wide suitable area for the potential distribution of each species in the neotropics, as promoted by climate variability. This work is important both in the scientific and technical communities since it allows for the direction of economic and environmental policies, creating early warning systems, mitigating the impacts of these pest species in the fruit and vegetable economies of Latin America and Panama, and contributing to improving the social economy and food security.

**Abstract:**

Climate variability has made us change our perspective on the study of insect pests and pest insects, focusing on preserving or maintaining efficient production systems in the world economy. The four species of the genus *Anastrepha* were selected for this study due to their colonization and expansion characteristics. Models of the potential distribution of these species are scarce in most neotropical countries, and there is a current and pressing demand to carry out this type of analysis in the face of the common scenarios of climate variability. We analyzed 370 presence records with statistical metrics and 16 bioclimatic variables. The MaxEnt method was used to evaluate the effect of the ENSO cycle on the potential distribution of the species *Anastrepha grandis* (Macquart), *Anastrepha serpetina* (Wiedemann), *Anastrepha obliqua* (Macquart), and *Anastrepha striata* (Schiner) as imported horticultural pests in the neotropics and Panama. A total of 3472 candidate models were obtained for each species, and the environmental variables with the greatest contribution to the final models were LST range and LST min for *A. grandis*, PRECIP range and PRECIP min for *A. serpentina*, LST range and LST min for *A. obliqua*, and LST min and LST max for *A. striata*. The percentage expansion of the range of *A. grandis* in all environmental scenarios was 26.46 and the contraction of the range was 30.80; the percentage expansion of the range of *A. serpentina* in all environmental scenarios was 3.15 and the contraction of the range was 28.49; the percentage expansion of the range of *A. obliqua* in all environmental scenarios was 5.71 and the contraction of the range was 3.40; and the percentage expansion of the range of *A. striata* in all environmental scenarios was 41.08 and the contraction of the range was 7.30, and we selected the best model, resulting in a wide distribution (suitable areas) of these species in the neotropics that was influenced by the variability of climatic events (El Niño, Neutral, and La Niña). Information is provided on the phytosanitary surveillance systems of the countries in areas where these species could be established, which is useful for defining policies and making decisions on integrated management plans according to sustainable agriculture.

## 1. Introduction

The fluctuations that are perceived in the climate (temperature, precipitation, and humidity) over relatively short periods are the cause of climate variability [1,2,3,4,5]. Previous studies have shown that climate variability affects physical and biological systems, and is of particular importance in cultivation areas, with changes in the frequency, intensity, spatial extent, duration, and timing of heavy rains and periods of drought during El Niño and La Niña phenomena. These variations can alter the behavior of ecosystems and cause the displacement and extinction of species [6,7,8,9]. El Niño–Southern Oscillation (ENSO) is a natural phenomenon involving fluctuating ocean temperatures in the equatorial Pacific [10]. Varying from 2 to 7 years, the ENSO cycle is the prevailing form of variability in the Pacific Ocean [6,11]. This variability causes uncertainties in productive agroecosystems (fruit and vegetables), in turn causing social and environmental stress that puts food security on the planet at risk [6,12]. It is expected that, in the next few years, climatic variability will cause changes in the geographic distribution of insects because of the reordering of climatic zones [13,14]. Therefore, the magnitudes of the impacts will be associated with climate variability. Regions of Latin America, South America, and the Caribbean are among the most vulnerable areas to climate variability, since most of the species that live there are endemic or restricted to a specific tropical ecosystem [15,16,17,18]. Most living beings are susceptible to the effects of climate variability due to specific physiologies and phenological qualities [19]. Poikilothermic organisms (ectotherms), such as insects, whose body temperatures vary according to the surrounding climate, are affected by unstable weather [20]. Climatic variations in temperature and precipitation can alter the ecological interactions of *Anastrepha* species, allowing species to colonize higher elevations and extend the ranges of their geographical distributions [21,22,23]. Two of the most commonly used methods to assess the effects of climatic variations in species distribution patterns have been the ecological niche (ENM) and potential distribution (SDM) models [24,25,26,27]. In addition to characterizing the environmental and geographic patterns of the species, these methods have been applied to multiple research questions in various areas of knowledge, including conservation, epidemiology and management of invasive species, eradication programs, and identification of cultivable areas [28,29,30].

The Tephritidae (Diptera) family consists of about 4700 species grouped into approximately 500 genera [31,32], distributed in temperate, tropical, and subtropical regions worldwide [33]. *Anastrepha* Schiner (Diptera: Tephritidae: Toxotrypa-nini) is the largest genus of Tephritidae from the Americas and includes more than 250 species distributed from the southern United States (Texas and Florida) to northern Argentina. [34,35]. At least seven species of the genus *Anastrepha* are considered economically important insects because they directly affect the production of cultivated fruit (*Mangifera indica* L. and *Citrus* sp.) and have a wide range of hosts [36]. Among them, *Anastrepha fraterculus* (Wiedemann 1830), *Anastrepha grandis* (Macquart 1846), *Anastrepha ludens* (Loew 1873), *Anastrepha obliqua* (Macquart 1835), *Anastrepha serpentina* (Wiedemann 1830), *Anastrepha striata* (Schiner 1868), and *Anastrepha suspensa* (Loew 1862) stand out. The species *A*. *fraterculus* is recognized as a cryptic species complex [37,38,39]. The damage caused by some species of the genus *Anastrepha* can be devastating to fruit and its commercialization; total losses of up to 90% of the crop have been documented in recent decades [40,41,42,43,44]. The potential distribution of fruit flies as part of the management of pest insects has been considered important in recent years, with the management of information bases, the use of models, and the implementation of new technologies in the fields of climate, species biodiversity, and better control planning in productive zones in regions of the United States [45], Europe [46,47], and Latin America [48,49,50].

Over the past decade, the world has seen at least 690 million people undernourished, 750 million people suffering from food insecurity, 2 billion people lacking access to safe and nutritious food, and 3 billion people unable to afford to pay the cost of a healthy diet [15]. Within the agricultural production sector, tropical fruit constitutes a relatively new group in the world trade of basic products, having acquired importance in the international market since 1970 due to advances in transportation, trade agreements, and changes in consumer fruit preferences [51]. With the declaration of the International Year of Fruits and Vegetables in 2021, healthy diets and lifestyles have been promoted among consumers in an environmentally sustainable way, recommending a minimum intake of 400 g of fruits and vegetables per person per day, which is only met in some parts of Asia and in countries with high incomes and good fruit availability [52,53]. In Panama, growing fruit has enormous potential for development due to the climate and soil conditions in various regions of the country, as well as the opportunities offered by international markets [54]. Of the total planted area (227,551 ha^−1^), as coverage of interest to the producer, 13% (29,581 ha) belongs to fruit trees [55].

Evaluating the effect of the ENSO cycle on the potential distribution of four species of the genus *Anastrepha*, of economic importance in horticulture, is of great relevance for Central America and Panama, and the scientific, economic, social, and environmental fields; it can provide information for decision-making in countries of the American continent that present the greatest vulnerability (exports and quarantines) for the establishment and distribution of each species. In addition, the information generated constitutes the basis for risk analysis and the implementation of preventive control and mitigation plans, which will reduce socioeconomic impacts in the potentially affected sectors of Latin America, Panama, and the world.

## 2. Materials and Methods

### 2.1. Study Area

The American neotropics, which extend between the Tropics of Cancer and Capricorn, was used as the study area [56,57]. This included the south of Mexico, all of Central America, the Caribbean, and a large part of South America, including the great Amazon jungle [58]. The natural landscape of these bioregions is made up of tropical forests, wetlands, savannahs, temperate grasslands, deserts, and Andean herbaceous formations [59,60,61,62]. Subsequently, layers were cut for the Republic of Panama, which is geographically located in the low northern latitudes (7°12′07″ and 9°38′46″ north latitude), between 77°09′24″ and 83°03′07″ western longitude, and characterized as the narrowest and longest country of the Central American isthmus. It has a land surface of 74,177.3 km^2^ (not including areas of continental water masses of 1,142,506.9 km^2^), is arranged in a west–east direction, and is bordered to the north by the Caribbean Sea, to the south by the Pacific Ocean, to the east by Colombia, and to the west by Costa Rica [63,64] (Figure 1).

### 2.2. Presence Records

As a model for this study, four species of the genus *Anastrepha* that are of horticultural importance in the neotropics and Panama were selected: *A*. *grandis*, *A*. *serpentina*, *A*. *obliqua*, and *A*. *striata* [65,66]. The various hosts, direct fruit damage, some polyphagous species, and distribution within the region were recorded [67]. Species presence records from the following sources were used: the Global Biodiversity Information Facility (GBIF; https://www.gbif.org/es/, accessed on 5 March 2022, Species Link (https://specieslink.net, accessed on 16 June 2022, Center for International Agricultural Bioscience (CABI; https://www.cabi.org/, accessed on 3 March 2022 and the National Plant Health Directorate of the Ministry of Agricultural Development (MIDA; https://mida.gob.pa/sanidad-vegetal, accessed on 10 March 2022. (Appendix A). All presence points were reviewed in Excel, eliminating duplicate records and those with coordinate errors, and using a shapefile of the neotropical polygon [68]. Once the database was formed, columns with the scientific name and longitude and latitude coordinates were ordered for each species. The data were refined using the thinning function with the spThin package in R [69], using a 30 km buffer to carry out the spatial thinning of records and taking the dispersal capacity reported for the genus *Anastrepha* as a reference [70,71,72]. Each final dataset was divided into presence records for calibration and evaluation (80% and 20%, respectively).

### 2.3. Climate Data

To characterize this phenomenon, the following specialized agencies were consulted for their records of climatic episodes occurring in the Pacific Ocean: NOAA in the United States (National Weather Service, Los Angeles, CA, USA, 2018), the Australian Government Meteorological Office (Bureau of Meteorology, Melbourne, Australia, 2018), and the Tokyo Climate Center in Japan (Japan Meteorological Agency, Tokyo, Japan, 2019). The information available from 2000 to 2019 was integrated; from the consensus of these three agencies, six episodes of La Niña, five of El Niño, and three Neutral episodes were obtained. Each of these episodes was characterized by four rasters (minimum, maximum, mean, and range), for which each of the environmental layers included the improved vegetation index (EVI, monthly Modis-Terra MOD 11C2v006), the temperature of the land surface (LST, monthly Modis-Terra MOD 11C3v006), near-real-time precipitation rate (NRTPR, 3 h TRMM 3B42RTv7), and normalized difference vegetation index (NDVI, monthly Modis-Terra MOD 11C2v006). From these, 16 environmental layers were obtained for each episode at a spatial resolution of 0.25° or 25 km at the equator [73,74,75].

With these variables, 8 datasets were built using the following criteria for their selection: all variables (set 1); analysis of the Pearson correlation coefficient to reduce collinearity (set 2) [76], where variables with a correlation value >|0.8| were removed using the corrplot package of the statistical software R 3.6.0 [77,78]; Jackknife analysis in MaxEnt to evaluate the individual contribution of variables without spatial autocorrelation to the models, which included variables that contribute ≥80% (set 3) [78]; variables with a variance inflation factor (VIF) <10 (set 4); all variables related to NDVI (set 5); all variables related to EVI (set 6); all variables related to LST (set 7); and all variables related to precipitation (set 8) [78,79,80,81] (Appendix A).

### 2.4. Construction of the Models

For each species, a calibration area was proposed for the model based on the spatial distribution of occurrences to define the environmental sampling area for the algorithm [82,83]. These areas were demarcated by taking the records of the presence of the species at the intersections of the polygons of the ecoregions proposed by the World Wide Fund for Nature (WWF) [56,84]. Using the MaxEnt program, which uses an automatic learning algorithm that combines Bayesian, maximum entropy, and statistical models, all possible models resulting from combinations of the variables were built, estimating the probability distribution of the maximum entropy of presences in a cell, determined according to the setting [85,86]. This probability distribution is the result of calculating the response curves, where suitability is described based on each of the model’s environmental variables [87]. All models were generated with the kuenm package in R version 1.1.5 using the method of exhaustive selection of variables [29,88,89], a sequence of 17 regularization multipliers (0.1, 0.2, 0.3, 0.4, 0.5, 0.6, 0.7, 0.8, 0.9, 1, 2, 3, 4, 5, 6, 8, and 10), 8 sets of variables, and a combination of entity classes (linear, quadratic, product, and their combination). Subsequently, the performance of the candidate models was evaluated with the following metrics: (a) statistical significance by partial receiver operating characteristic curve (pROC with 500 iterations and 50 percent of the data for bootstrapping); (b) evaluation of the omission rate in the test presence records, selecting models with results lower than 5%; and (c) selection of models with less complexity to the Akaike information criterion corrected for small samples, and the difference between the pretender model and the outstanding model (AICc and ∆AICc, respectively). The models were evaluated by selecting those that were statistically significant (criterion a) and reducing the number of models by incorporating only those that executed the omission rate (criterion b). Among the significant and low omission candidate models, models with a delta less than 2 (criterion c) were selected [76]. Based on the results, each final model was built from 10 repetitions, and individual response curves were obtained. The type of output of the final model was selected (extrapolation, free extrapolation, and without extrapolation and subjection) using Jackknife analysis of the three variables with the greatest contribution to the models and graphing them in a three-dimensional space. In this environmental space, the location of the points of occurrence is compared with the environmental space of area M [77,81]. We reclassified the maps to binary by a threshold that omits all regions with habitat suitability below the suitability values for the lowest 10% of occurrence records [50] using the reclassify to binary function in SDMToolbox V. 2.4.

### 2.5. Analysis of Geographic Space

Each final model by species was projected to the geographic space of the area of the neotropics, and later the map was cut to the area of Panama, estimating the potential distribution of each species in the different average conditions investigated (Neutral, El Niño, and La Niña). These projections were reclassified as binary (1 = suitable; 0 = not suitable) [90,91,92], calculating a threshold for each species that omitted all regions with habitat suitability below the suitability values for the lower 10% of occurrence records. Additionally, we implemented a mobility-oriented parity (MOP) analysis that compared the environmental similarity of the projection area with the model calibration area to identify areas where there was strict extrapolation [93,94,95,96,97,98].

## 3. Results

A total of 370 presence records were obtained for the four *Anastrepha* species, in which *A*. *grandis* had the lowest number of records (39), distributed between Panama, Colombia, Ecuador, Venezuela, Peru, Bolivia, Brazil, Paraguay, Uruguay, and northern Argentina. *Anastrepha serpentina* had 88 records distributed among Mexico, Belize, El Salvador, Guatemala, Costa Rica, Panama, Colombia, Ecuador, Venezuela, French Guyana, Peru, Brazil, Bolivia, Paraguay, and northern Argentina. *Anastrepha obliqua* had 93 records distributed in Mexico, Guatemala, Belize, El Salvador, Nicaragua, Costa Rica, Panama, Colombia, Venezuela, Peru, Ecuador, Suriname, French Guiana, Brazil, Bolivia, Paraguay, and northern Argentina. Finally, *A*. *striata* had 150 records, mainly distributed in Mexico, Belize, Honduras, Guatemala, Nicaragua, Costa Rica, Panama, Colombia, Ecuador, Peru, French Guyana, Suriname, Bolivia, Brazil, and Paraguay. A total of 3472 candidate models were obtained for each *Anastrepha* sp. Comparing the locations of the presence records in the environmental space and the environmental space available in the calibration area (M) and the projection area for the four species, the records were dispersedly distributed in the environmental space of the M. This result allowed the selection of all species in the models with output and extrapolation, considering the methodology proposed by Owens et al. [93] (Appendix A). After analyzing all candidate models, the best of each species was selected according to the criteria outlined in the Materials and Methods section (Table 1).

The environmental variables with the greatest contributions to the final models were LST range (32%) and LST min (28%) for *A*. *grandis*, PRECIP range (58%) and PRECIP min (18%) for *A*. *serpentina*, LST range (45%) and LST min (28%) for *A*. *obliqua*, and LST min (47%) and LST max (24%) for *A*. *striata*.

### 3.1. Potential Distribution in the Neotropics

Under El Niño, Neutral, and La Niña conditions, the models for the four species showed wide areas with environmentally suitable conditions (adequate for their establishment) in coastal, savannah, montane, temperate, and tropical forest zones. For *A*. *grandis*, the potential distribution models in El Niño (Figure 2A) and La Niña (Figure 2C) events were very similar, with suitable areas in northern and southeastern to western Mexico, Guatemala, Belize, northwestern Honduras and Nicaragua, southwestern Guatemala, Costa Rica, the Atlantic zone and western Panama, northeastern Colombia and Ecuador, southeastern Venezuela, northeastern Guyana, northeastern Peru, northern Bolivia, much of the northern and southern Amazon in Brazil, southeastern Paraguay, Uruguay, northern and southern Argentina, and southern Chile. For Neutral events (Figure 2B), a larger suitable area was evident in northern Mexico, the northern Amazon in Brazil, central Uruguay and Paraguay, northern Argentina, and several countries in Central America. For *A*. *serpentina*, the El Niño (Figure 3A) and La Niña (Figure 3C) episodes have almost the same pattern, where an increase in the suitable areas was observed in the northeastern areas of Mexico during the La Niña event and a decrease from the ideal area in southern Guatemala, northern Honduras, and El Salvador, and an increase in Costa Rica and Panama. In South America, there was an increase in the ideal areas in a large part of Colombia, Venezuela, Ecuador, Brazil, Bolivia, Paraguay, and northern Argentina and a reduction in the suitable area in La Niña in southern Brazil, northern Argentina, and northwestern Chile. In the Neutral event, there was a larger suitable area in the entire neotropical region except in the central and western areas of Mexico, a small area of northwestern and southern Colombia, east-central Brazil, the coastal area of Peru and Chile, and northeastern and southern Argentina (Figure 3B).

For *A*. *obliqua*, in the El Niño and La Niña events, the ideal area increases in a large part of the neotropics, in Mexico, Central America, and a large part of the South American territory (Figure 4A,C). In Neutral events, it maintains a pattern similar to that of El Niño and La Niña, except for the increase in the suitable area that is observed in central Mexico (Figure 4B). Lastly, for *A*. *striata*, the El Niño and La Niña events (Figure 5A,C) presented a more suitable area in Mexico, Guatemala, Belize, El Salvador, Nicaragua, Costa Rica, Panama, Colombia, Venezuela, central Brazil, Argentina, Paraguay, Uruguay, and part of Chile. In Neutral events, the increase was maintained in central Brazil and southern Argentina (Figure 5B).

### 3.2. Potential Distribution in Panama

In the case of Panama, the potential distribution with suitable areas for *A*. *grandis* in the El Niño event was in the eastern (Darién, Panama) and northern parts of the country (Colon, Coclé, Veraguas, and Bocas del Toro). In Neutral events, the ideal area included the provinces of Darién and Panama to the east, Panama West, the Atlantic area of Colon, Veraguas, and Bocas del Toro, and the central areas of Herrera, Coclé, Veraguas, and Chiriquí. The ideal areas remained in eastern and western Panama, Darién, the Atlantic zone of Colon, Veraguas, and Bocas del Toro, and the Pacific zone of Veraguas and central Chiriquí (Figure 6A). For *A*. *serpentina* and *A*. *obliqua*, the ideal areas covered all of Panama in all events (Figure 6B,C). For *A*. *striata*, in El Niño, the ideal area covered almost all of Panama, except for isolated points in Colon and Bocas del Toro. In Neutral events, the ideal areas were located in the central regions of the provinces of Darién, Panama, Coclé, Veraguas, Chiriquí, and Bocas del Toro, and to the south in Los Santos. In La Niña events, the ideal areas were eastern Darién, central Panama, and the provinces of Coclé, Herrera, Los Santos, Veraguas, Chiriquí, and Bocas del Toro (Figure 6D).

### 3.3. Movement-Oriented Parity Analysis (MOP)

When identifying the risk of extrapolation of El Niño, Neutral, and La Niña conditions, all species presented similar patterns, where most of the projection area presented environmental similarities with the calibration area (Figure 2D–F). Specifically, for *A*. *grandis*, in El Niño events, there was a risk of extrapolation in northern and central Mexico, southern Peru, northern Chile, northeastern Argentina, and east-central Brazil. In Neutral events, there was a risk of extrapolation in central Mexico, southern Peru, northern Chile, northeastern Argentina, and Brazil. For La Niña, we identified areas of extrapolation risk in western and eastern Mexico, east-central Brazil, southern Peru, northern Chile, and northeastern Argentina. For *A*. *serpentina*, in El Niño events, there was no risk of extrapolation. For Neutral events, there was a risk of extrapolation in west-central Mexico, southern Peru, and parts of southern Argentina. For La Niña events, there was a risk of extrapolation in southwestern Mexico, northern and southern Guatemala, northern El Salvador, northern Belize, northern Colombia and Venezuela, and east-central Brazil (Figure 3D–F). For *A*. *obliqua*, in El Niño events, there was a risk of extrapolation in west-central Mexico, southern Peru, northern Chile, northern Argentina, and east-central Brazil. For Neutral events, there was a risk of extrapolation in central Mexico and northern Chile, and for La Niña events, there was a risk of extrapolation in a large part of Mexico, northern Venezuela, a large part of east-central and southern Brazil, Argentina, Paraguay, Uruguay, southern Peru and Bolivia, and Chile (Figure 4D–F). For *A*. *striata*, in El Niño events, there was a risk of extrapolation in northwestern Mexico, northern Venezuela, east-central Brazil, southern Peru, and on the coasts of Chile and Argentina. In Neutral events, there was a risk of extrapolation at only a few points in northern Mexico, northern Chile, and northern Argentina. In La Niña events, there was a risk of extrapolation in a large part of Mexico, northern Venezuela, east-central and southern Brazil, Paraguay, Uruguay, Argentina, southern Peru and Bolivia, and Chile. In the remaining points not mentioned by events, there was no risk of extrapolation (Figure 5D–F).

In the MOP analysis, in Panama, there was no risk of extrapolation in most of the projected models of the species, except for *A*. *grandis* in Neutral events in Darién Province (Figure 2E), *A*. *serpentina* in El Niño events in Darién Province (Figure 3D), and La Niña in the provinces of Herrera and Los Santos (Figure 3E).

### 3.4. Analysis of Changes in Environmental Suitability in Different Scenarios

For *A*. *grandis*, when comparing the Neutral vs. El Niño scenarios, there was a 14.72% expansion range to northern Mexico, southwestern Brazil and Colombia, southern Peru, and a large part of Chile. In addition, 61.53% of the area was shown as an area of no change (occupied area in both scenarios), mainly for the Pacific zone, southern Peru, Chile, and Argentina. In the comparison of Neutral vs. La Niña events, there was a 16.47% contraction in northern Mexico, southeastern Venezuela, east-central Brazil, southern Bolivia, and northern Paraguay and Argentina, with a 59.38% chance of no change in the occupied area (Figure 7A). For *A*. *serpentina* (Figure 7B), a 14.11% contraction range was observed in Neutral vs. El Niño events in northern Mexico, northeastern Venezuela, east-central Brazil, central Bolivia, Paraguay, and Argentina, with a 74.57% chance of no change in the occupied area. In Neutral vs. La Niña events, there was a 14.38% contraction range in northern Mexico, central Colombia and Peru, and southeastern Brazil, Uruguay, and Argentina, with a 74.29% chance of no change in the occupied area. In *A*. *obliqua* (Figure 7C), in Neutral vs. El Niño events, we observed a 9.77% unoccupied area in parts of Colombia, Ecuador, Peru, Bolivia, Chile, and Argentina, and an 86.4% chance of no change in the occupied area. In Neutral vs. La Niña events, there was an 8.04% unoccupied area in Colombia, Ecuador, Peru, Bolivia, Chile, Argentina, and southern Brazil, with an 86.59% chance of no change in the occupied area. For *A*. *striata* (Figure 7D), in Neutral vs. El Niño events, we observed an 18.91% range expansion in a large part of Mexico, northern Guatemala and Belize, and central Brazil, in addition to a 66.84% chance of no change in the occupied area. In Neutral vs. La Niña events, there was a 22.17% expansion range in Mexico, northern Guatemala and Belize, and central Brazil, and a 68.44% chance of no change in the occupied area (Table 2).

## 4. Discussion

In recent years, the study of new technologies has allowed strategic plans to be made within productive systems, where the management of information on biodiversity is important, since these elements allow prevention and control plans to be organized in various countries, improving food security in society [99,100,101]. The results of this study indicate that the studied species of the genus *Anastrepha* can increase in the native areas at the neotropical level. In ENSO events (El Niño, Neutral, and La Niña), the information generated is important due to the limited research on species of this genus [102,103]. For *A*. *grandis*, there are reports of its presence from eastern Panama to a large part of South America, for which it is important to know the places with ideal environmental conditions for its distribution to the rest of Central America [104]. In El Niño and La Niña events, the geographic spaces with ideal environmental conditions for potential distribution were similar, where the LST variable (range and min) was shown to be the most important in the models. This result could be related to the stability of soil moisture and temperature in tropical forested areas at the time of oviposition, hatching, and the availability of fruit hosts and wild host plants [104,105]. The results of this study showed that areas with adequate conditions for these species of fruit flies corresponded to localities with temperate and sub-humid climates in Central America and northern Mexico. Therefore, the government entities in charge of the control of fruit flies and other pest insect species should take measures that allow the early arrest of these species and thus reduce the risk of the establishment of reproductive populations [104]. During Neutral events, the model showed an increase in geographic spaces with adequate conditions in the center of the Amazon. This could be related to fluctuations in average and low temperatures in this region, which benefit the biological processes of hatching and larval development of *A*. *grandis* [105]. In Panama, this species could inhabit tropical forests, where they could expand toward suitable productive zones, according to the results [104,105,106].

The results of the MOP analysis for *A*. *grandis* showed a high environmental similarity between the calibration and projection areas in the different episodes that could favor the latitudinal and altitudinal expansion of the geographic distribution of the species. This pattern is potentially due to high climate variability in the average temperatures in the regions of Central America and central South America where there is a greater concentration of tropical forests, and with little extrapolation irrigation in central Mexico and Panama due to their arid zones [106,107,108]. When comparing Neutral vs. El Niño events, the potential distribution of *A*. *grandis* could expand because the biology of the species responds well to average temperatures in northern Central America, where there are savannahs and temperate forests, similar to South American regions with temperate and semi-arid savannah forests in which mild temperatures and little rainfall favor the expansion of this species [109,110]. The changes observed in Neutral vs. La Niña events would cause a contraction in the species possibly associated with high temperatures in northern Mexico, due to its dry areas and low humidity, affecting the species’ reproduction in the same way as in South America due to the semi-arid conditions with high temperatures in western Amazonia and semi-arid conditions in Argentina [111,112].

For *A*. *serpentina*, the potential distribution in El Niño and La Niña episodes could be interpreted according to the model based on the climate variable PRECIP (range and min), where in arid areas with little rainfall, such as northern Mexico and the Southern Cone countries, there is little probability of establishing this species [113,114,115]. In Neutral episodes, the adequate area could increase due to the average rainfall in the neotropics since the biological processes associated with the species adapt better with adequate field humidity [114]. When projecting MOP analyses in Neutral and El Niño events, the environmental conditions could be similar since by maintaining the average rainfall regime in the neotropics, the species increases its distribution [113]. In La Niña events, the environmental similarity is lower since the increase or variation in the amount of rain in east-central Brazil affects the reproduction of the species due to the death of males [114,115,116]. In Panama, suitable zones with environmental similarities are high, which would not affect the distribution patterns of the species, where the natural landscape and orography probably maintain good humidity conditions in the environment [117]. When comparing the Neutral vs. El Niño events, the potential distribution could contract in the north of Mexico, part of the coastal areas of the central Pacific, and the Southern Cone due to their arid conditions or lack of rainfall, affecting the hatching and lifespan of the species [117,118]. When comparing Neutral and La Niña events, there could be a contraction in the distribution due to the low rainfall produced on the Mexican Atlantic coast and a large part of the Southern Cone, which affects egg oviposition and pupal development due to a lack of humidity [40,118,119]. For *A*. *obliqua*, one of the most polyphagous species of this genus, the climate variable LST (range and min), according to the model, could influence the potential distribution in El Niño and La Niña events, since it adapts to a wide range of temperatures, colonizing almost the entire neotropics, with countless hosts, limited only by high temperatures and competition between species and predators, which is why their development can be delayed or accelerated [120]. At an ecological level, the areas with high elevation, as well as cold climates, reduce their distribution, such as the Cordillera of Mexico, the Andes in South America, and the coldest parts of the Southern Cone [121,122]. In Neutral events, the possible extension of suitable distribution areas in northern Central America could be due to the stability in temperatures and the greater range of fruit hosts, which benefit the reproduction of this species [122]. Observing the MOP analysis, in El Niño, Neutral, and La Niña events, the areas with the greatest environmental similarity between the calibration and projection areas were from Neutral events, which favor the greater distribution of the species due to its characteristics of adaptation to variations in temperatures [120]. In El Niño and La Niña events, the environmental similarity is reduced in parts of Mexico due to their hot, dry, and very humid climates, in eastern and southern Brazil due to dry climates, and in Argentina due to its cold climate, which harms the oviposition of the eggs of this species [120,122]. In Panama, the area of potential distribution and environmental similarity presents adequate zones in its geography, which could promote reproduction (variety of hosts and ideal conditions) in the country due to its suitable temperature (30 °C) and humidity (98%) [123,124]. The changes observed for Neutral vs. El Niño events showed that the potential distribution of the species would not include the Andes or Argentina; this could be related to low temperatures and the few natural hosts present in these areas [40,45]. In the comparison of Neutral vs. La Niña, both presented the same pattern as El Niño, with their potential distribution areas not including those with cold climates, affecting all reproduction patterns [120,121,122,123,124,125].

For *A*. *striata*, better known as the guava fly, the potential distribution could be influenced by the climate variable LST (min and max), according to the projected model, since El Niño and La Niña events present similar distribution patterns. We estimate that the species adapts to various ranges of temperatures between 18 °C and 30 °C and good soil moisture throughout the neotropics, except for the mountainous regions of the Andes and the wetlands of Brazil [126,127,128]. In Neutral events, the areas of potential distribution are reduced in central Mexico due to its desert areas with high temperatures and in northern Brazil, in the jungle areas of the Amazon with high temperatures and high humidity, which affect the hatching and death of adults [126]. In Panama, for El Niño events, the area of potential distribution is wide, which tells us that the species adapts to several not very humid tropical temperature ranges. In Neutral and La Niña events, the areas of potential distribution are limited only to coastal areas due to their high temperatures and low humidity, which harms the reproduction of the species [128]. When reviewing the MOP maps in Neutral events, there is a high environmental similarity compared to the area of distribution. The species may adapt to variations in climate, temperatures, and tropical humidity, except in very high areas [126,127]. During El Niño and La Niña, there is a high risk of extrapolation of the species in northwestern Mexico and southern Brazil, Uruguay, Paraguay, Chile, and Argentina, since the areas of environmental similarity are low, and the species cannot be present due to the climatic conditions of adverse temperatures for its reproduction [102]. The changes observed for Neutral vs. El Niño and Neutral vs. La Niña events could present an increase in the expansion areas in central Mexico due to its semi-warm and tropical conditions, places where temperatures are high, such as desert areas and tropical coastal areas, dry areas in Venezuela, and tropical forests in east-central Brazil, which demonstrates the great adaptability of *A. striata* in its adult phase at high temperatures [126,127]. The same happens in Panama, where the range of expansion is toward the tropical mountain areas on the Atlantic and Pacific coasts, where temperatures and humidity are high, which benefits the reproduction of the species during its dispersal phase in search of hosts [20,126,127,128,129,130].

## 5. Conclusions

There is evidence that indicates that changes in temperature and rainfall induced by the ENSO cycle cause changes in the potential distribution of insect species by influencing the community structure and population dynamics. Our results show that the climatic variability caused by the event in turn causes changes in the expansion of *Anastrepha* spp. of horticultural importance, occupying an extensive area where favorable and unfavorable zones can be altered in a varied and complex way in the neotropical region. The anomalies in precipitation and temperature induced by the El Niño and La Niña phenomena generate changes in the potential distribution of species in the region, and an increase or reduction in the environmentally suitable zone is projected. The applied modeling methodology is efficient and makes the simulation of the system more accessible, indicating areas of greater risk. Information is provided for use in phytosanitary surveillance systems of the countries in areas where these fruit fly pests could be established, to define policy and decision-making on integrated management plans according to sustainable agriculture principles.

## Figures and Tables

**Figure 1 insects-14-00714-f001:**
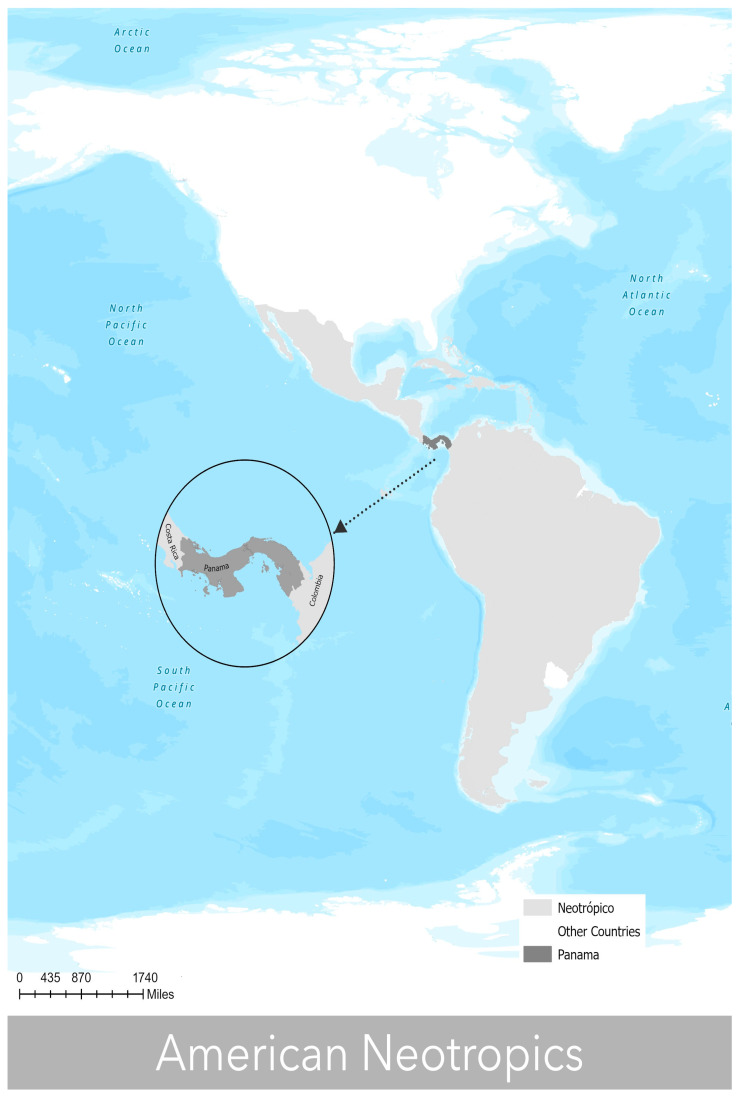
Study area map of potential distribution models.

**Figure 2 insects-14-00714-f002:**
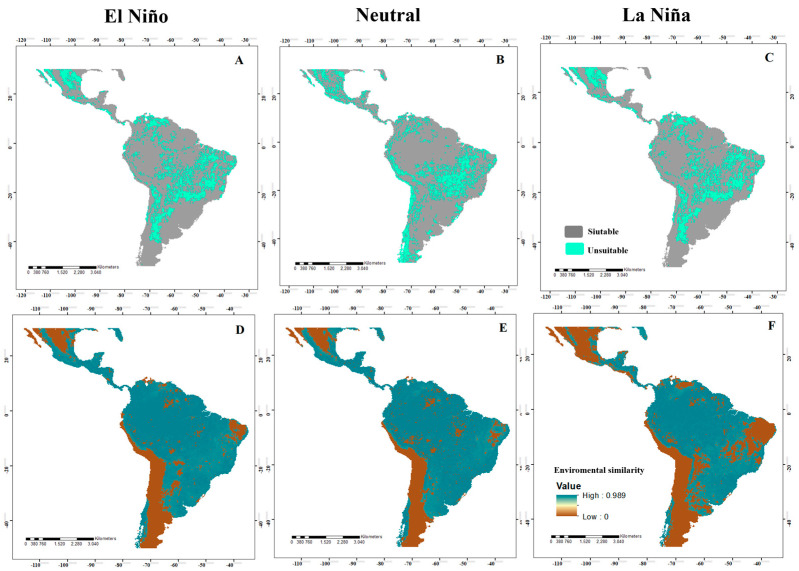
Potential distribution of *Anastrepha grandis* in the neotropics for El Niño, Neutral, and La Niña events ((**A**–**C**), respectively), where high suitability values (suitable areas) are represented in gray and low suitability (unsuitable areas) are in green. In addition, various maps are shown analyzing the MOP in the different El Niño, Neutral, and La Niña events ((**D**–**F**), respectively), where high environmental similarity values are represented in blue and low environmental similarity in brown.

**Figure 3 insects-14-00714-f003:**
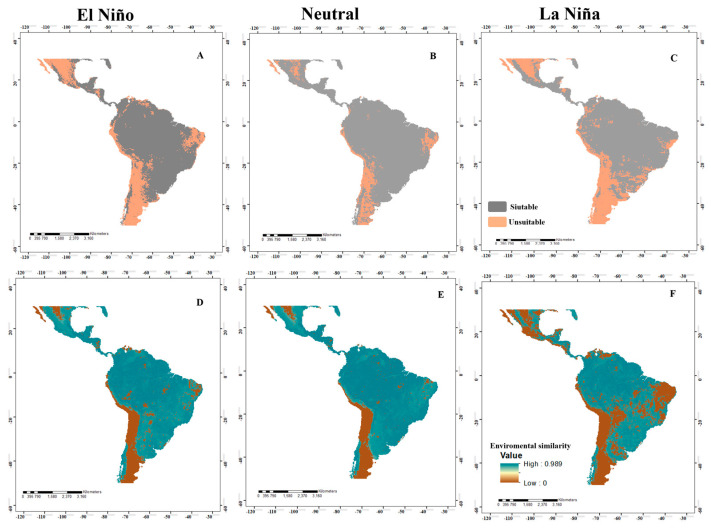
Potential distribution of *Anastrepha serpentina* in the neotropics for El Niño, Neutral, and La Niña events ((**A**–**C**), respectively), where high suitability values (suitable areas) are represented in gray and low suitability (unsuitable areas) are in orange. In addition, various maps are shown analyzing the MOP in the different El Niño, Neutral, and La Niña events ((**D**–**F**), respectively), where high environmental similarity values are represented in blue and low environmental similarity in brown.

**Figure 4 insects-14-00714-f004:**
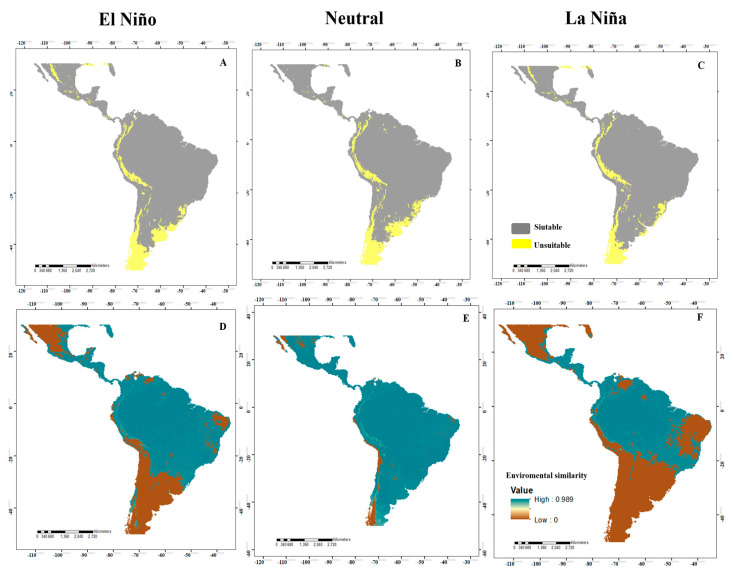
Potential distribution of *Anastrepha obliqua* in the neotropics for El Niño, Neutral, and La Niña events ((**A**–**C**), respectively), where high suitability values (suitable areas) are represented in gray and low suitability (unsuitable areas) are in yellow. In addition, various maps are shown analyzing the MOP in the different El Niño, Neutral, and La Niña events ((**D**–**F**), respectively), where high environmental similarity values are represented in blue and low environmental similarity in brown.

**Figure 5 insects-14-00714-f005:**
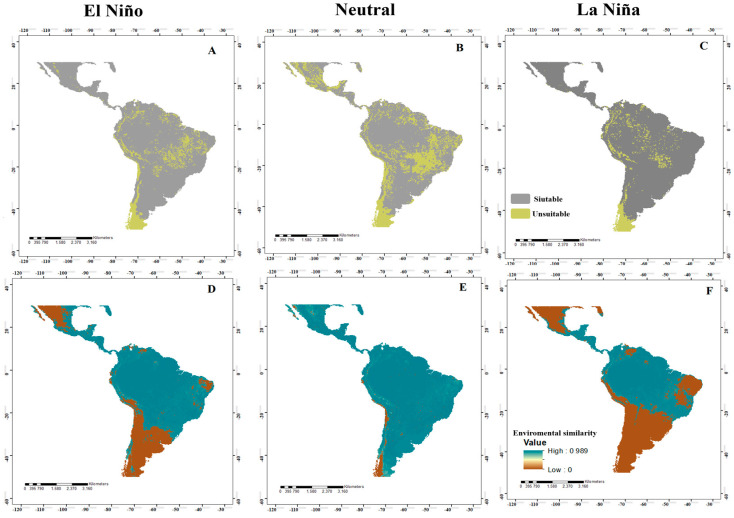
Potential distribution of *Anastrepha striata* in the neotropics for El Niño, Neutral, and La Niña events ((**A**–**C**), respectively), where high suitability values (suitable areas) are represented in gray and low suitability (unsuitable areas) are in green. In addition, various maps are shown analyzing the MOP in the different El Niño, Neutral, and La Niña events ((**D**–**F**), respectively), where high environmental similarity values are represented in blue and low environmental similarity in brown.

**Figure 6 insects-14-00714-f006:**
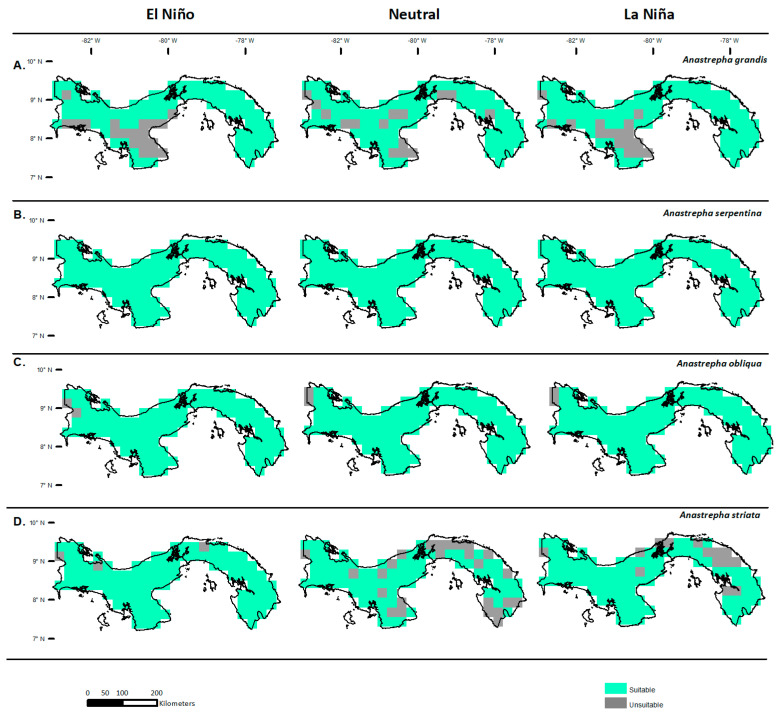
Potential distribution of *Anastrepha* spp. in Panama for the El Niño, Neutral, and La Niña events: *A*. *grandis* (**A**), *A*. *serpentina* (**B**), *A*. *obliqua* (**C**), and *A*. *striata* (**D**). The suitable areas are shown in green, and the unsuitable areas are represented in gray.

**Figure 7 insects-14-00714-f007:**
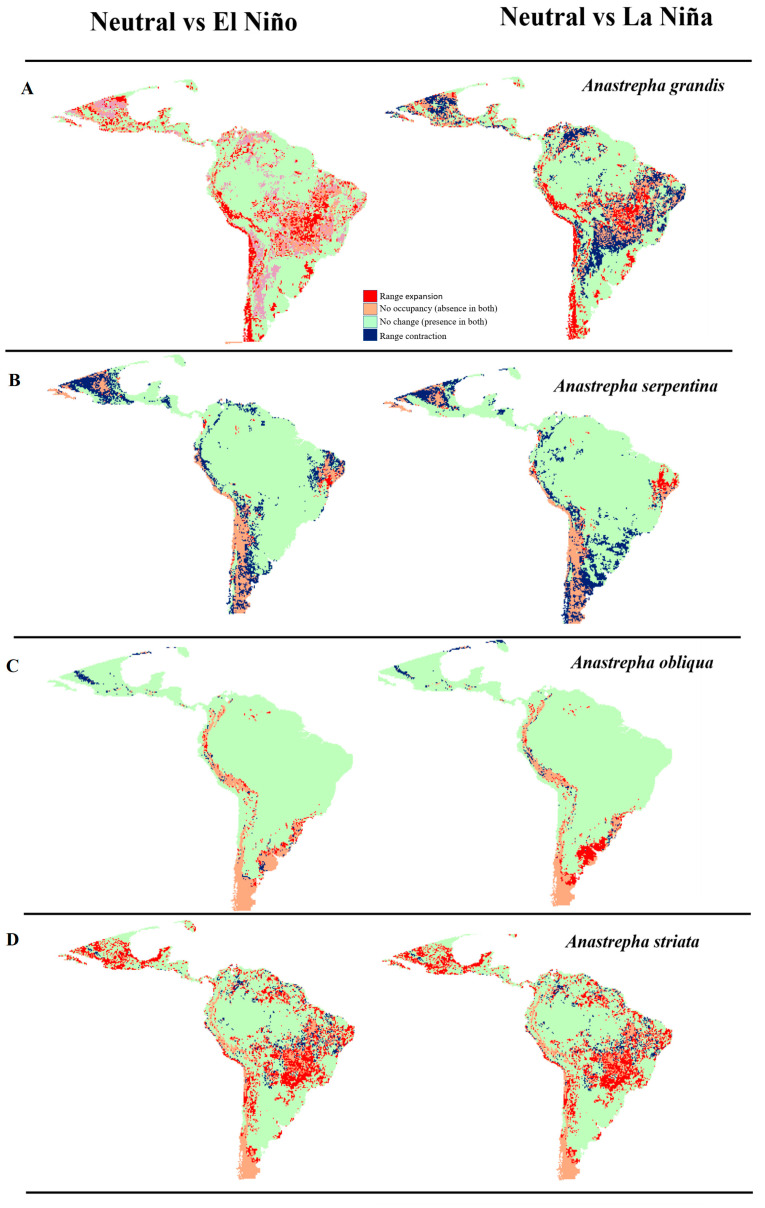
Analysis of changes in the ranges of expansion, no occupation, no change, and contraction range in the different scenarios for each species ((**A**)–*A. grandis*, (**B**)–*A. serpentina*, (**C**)–*A. obliqua* and (**D**)–*A. striata*) in the comparison of Neutral vs. El Niño and Neutral vs. La Niña events.

**Table 1 insects-14-00714-t001:** Models selected based on defined criteria with partial ROC statistical significance, skip rate performance, and AICc complexity.

Species	Records	Train	Selected Model	Partial ROC	Omission Rate (<5%)	AICc	∆AICc
*Anastrepha grandis*	39	32	M_0.9_F_t_Set7	0	0	392.882	0.000
*Anastrepha serpentina*	88	70	M_0.4_F_lq_Set8	0	0	1017.849	0.000
*Anastrepha obliqua*	93	74	M_4_F_l_Set7	0	0	1080.475	0.000
*Anastrepha striata*	150	125	M_3_F_lt_Set7	0	0	1854.480	0.000

**Table 2 insects-14-00714-t002:** Suitability percentages in the neotropics for each species of *Anastrepha* in the comparison of Neutral vs. El Niño and Neutral vs. La Niña events.

	*A. grandis* Neutral vs. Niño	*A. grandis* Neutral vs. Niña	*A. serpentina* Neutral vs. Niño	*A. serpentina* Neutral vs. Niña	*A. obliqua* Neutral vs. Niño	*A. obliqua* Neutral vs. Niña	*A. striata* Neutral vs. Niño	*A. striata* Neutral vs. Niña
Range expansion	14.72	11.74	1.17	1.98	1.99	3.72	18.91	22.17
No occupancy	9.41	12.38	10.14	9.33	9.77	8.04	9.82	6.56
No change	61.53	59.38	74.57	74.29	86.4	86.59	66.84	68.44
Range contraction	14.33	16.47	14.11	14.38	1.77	1.63	4.41	2.89

## Data Availability

All data are contained in the article and the Appendix A.

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
