# Peer review of "Evaluation of the Effect of the ENSO Cycle on the Distribution Potential of the Genus Anastrepha of Horticultural Importance in the Neotropics and Panama"

_insects, 2023, doi:10.3390/insects14080714_

Round 1

Reviewer 1 Report

This paper used Maxent to model the influence of the ENSO cycle on important pests in tropical regions. The introduction is thorough and the methods and justification for modelled is described in detail.

There are just some minor comments/suggestions below.

Line 14. Consider for clarity “The genus Anastrepha (Diptera: Tephritidae) are a group of fruit flies that contain significant insects (pests) of agricultural importance by altering the quality and reducing yield of fruit and vegetable production. Thus, projecting great uncertainty about safety due to quarantine restrictions around the world.”

Line 20 “these limiting insects”, consider changing “insects” to “pests”. Please be consistent to highlight the studies importance as at Line 24 you use “limiting species (pests)”. Then it is very confusing at line 27 “insect pests and limiting insects”

Line 29 “limit insects of quarantine importance in most countries of the world, significantly reducing the export of tropical fruit”. Is it meant to be “interception of insects of quarantine importance”

Line 36  “of importance in horticulture” consider changing to  “as import horticultural pests….”

Line 43 climate variability is fluctuations in variables, and should consider this revised sentence “Climate variability refers to the fluctuations in a climate variable (i.e. temperature, precipitation, and humidity) around their average values.” Delete “which is more alarming every day in the world”.

Line 49-52consider changing  “Previous studies have shown that climate variability affects physical and biological systems, and is of particular importance in cultivation areas, with changes in frequency, intensity, spatial extent, duration, and timing of, heavy rains, and periods of drought during El Niño and La Niña phenomena. These variations can alter the behavior of ecosystems and causing the displacement and extinction of species”.

Line 61 insert “,” after variability.

Line 141 “from various” change to “from the following”

Line 491 and also at Line 501 insert “A. striata” and delete “of the species”. As it reminds the reader this paragraph is still talking about the same species mentioned at the start of the paragraph.

Line 514-517 consider changing to “Information is provided for use in phytosanitary surveillance systems of these countries in areas where these fruit fly pests could be established, to define policy and decision making on integrated management plans according to sustainable agriculture principles.”

References

Can the reference [6] be used more often in the manuscript, and if possible use this reference below

“IPCC, 2022: Climate Change 2022: Impacts, Adaptation, and Vulnerability. Contribution of Working Group II to the Sixth Assessment Report of the Intergovernmental Panel on Climate Change [H.-O. Pörtner, D.C. Roberts, M. Tignor, E.S. Poloczanska, K. Mintenbeck, A. Alegría, M. Craig, S. Langsdorf, S. Löschke, V. Möller, A. Okem, B. Rama (eds.)]. Cambridge University Press. Cambridge University Press, Cambridge, UK and New York, NY, USA, 3056 pp., doi:10.1017/9781009325844”

Supplementary

Table sS2

It is mentioned in the text Lines 170-178 what each set means, though the variables used in sets 5,6,7 and 8 are not listed for each species and should be included given this is a supplementary, otherwise the table is almost not needed, as the variables in set 2,3 and 4 for each species could be combined into one table for all species modelled.

english is mostly fine, just be wary of using "insect" in place of "pests" and vice versa.

Author Response

Comments and Suggestions for Authors

This paper used Maxent to model the influence of the ENSO cycle on important pests in tropical regions. The introduction is thorough and the methods and justification for modelled is described in detail.

There are just some minor comments/suggestions below.

Line 14. Consider for clarity “The genus Anastrepha (Diptera: Tephritidae) are a group of fruit flies that contain significant insects (pests) of agricultural importance by altering the quality and reducing yield of fruit and vegetable production. Thus, projecting great uncertainty about safety due to quarantine restrictions around the world.”

Thanks for the suggestions we change the paragraph following the recommendation

Line 20 “these limiting insects”, consider changing “insects” to “pests”. Please be consistent to highlight the studies importance as at Line 24 you use “limiting species (pests)”. Then it is very confusing at line 27 “insect pests and limiting insects”

Thanks for the recommendation, we changed the limiting term for plague throughout the manuscript.

Line 29 “limit insects of quarantine importance in most countries of the world, significantly reducing the export of tropical fruit”. Is it meant to be “interception of insects of quarantine importance”

Thanks for the suggestion We eliminated the term of insects of quarantine importance to avoid confusion

Line 36  “of importance in horticulture” consider changing to  “as import horticultural pests….”

Thanks, done

Line 43 climate variability is fluctuations in variables, and should consider this revised sentence “Climate variability refers to the fluctuations in a climate variable (i.e. temperature, precipitation, and humidity) around their average values.” Delete “which is more alarming every day in the world”.

Thank you very much, we deleted the text part as suggested

Line 49-52consider changing  “Previous studies have shown that climate variability affects physical and biological systems, and is of particular importance in cultivation areas, with changes in frequency, intensity, spatial extent, duration, and timing of, heavy rains, and periods of drought during El Niño and La Niña phenomena. These variations can alter the behavior of ecosystems and causing the displacement and extinction of species”.

Thanks for the suggestion, we changed the wording of the paragraph following your recommendation

Line 61 insert “,” after variability.

Thanks done

Line 141 “from various” change to “from the following”

Thanks done

Line 491 and also at Line 501 insert “A. striata” and delete “of the species”. As it reminds the reader this paragraph is still talking about the same species mentioned at the start of the paragraph.

Thanks done

Line 514-517 consider changing to “Information is provided for use in phytosanitary surveillance systems of these countries in areas where these fruit fly pests could be established, to define policy and decision making on integrated management plans according to sustainable agriculture principles.”

Thanks for the suggestion, we changed the wording of the paragraph following your recommendation

References

Can the reference [6] be used more often in the manuscript, and if possible use this reference below

“IPCC, 2022: Climate Change 2022: Impacts, Adaptation, and Vulnerability. Contribution of Working Group II to the Sixth Assessment Report of the Intergovernmental Panel on Climate Change [H.-O. Pörtner, D.C. Roberts, M. Tignor, E.S. Poloczanska, K. Mintenbeck, A. Alegría, M. Craig, S. Langsdorf, S. Löschke, V. Möller, A. Okem, B. Rama (eds.)]. Cambridge University Press. Cambridge University Press, Cambridge, UK and New York, NY, USA, 3056 pp., doi:10.1017/9781009325844”

Thanks for the suggestion, we include reference 6 more frequently in the introduction section

Supplementary

Table sS2 

It is mentioned in the text Lines 170-178 what each set means, though the variables used in sets 5,6,7 and 8 are not listed for each species and should be included given this is a supplementary, otherwise the table is almost not needed, as the variables in set 2,3 and 4 for each species could be combined into one table for all species modelled.

Thank you, we modified supplement 2 following the suggestion

Comments on the Quality of English Language

english is mostly fine, just be wary of using "insect" in place of "pests" and vice versa.

Thanks done

Reviewer 2 Report

The manuscript is a significant contribution involving variability of climatic events (El Niño, Neutral, and La Niña) via maxent model for the habitat suitability of economically important pests. Authors have done well with reliable results for scientific community. I have some suggestions for improvement.

Miner improvement suggested.

Author Response

Comments for authors

The manuscript is a significant contribution involving variability of climatic events (El Niño, Neutral, and La Niña) via maxent model for the habitat suitability of economically important pests. Authors have done well with reliable results for scientific community. I have some suggestions for improvement.

Title

Good

Abstract:

This part needs modifications as half of abstract in current form only introducing the topic. It must be with slight introduction, methodology, results then conclusion. So modify this part under the light of below given suggestions.

1.Line 29-30 is too long , make it shorter to remove grammatical errors

Thanks for the suggestions, we removed some of the text following the suggestion

2.You may add finally selected variables along with their contribution, AUC and Regular multiplier etc. Values of matics , finally selected model criteria etc.

3.Calculate areas of contraction and expansion for all in kilometers or percentages for all events and add here in abstract part.

4.In the last lines you must add some suggestions for policy makers, researchers Gov. agencies for possible measure which should be under taken.

5.You must also mentions areas of species expansion and contraction. Where there is risk of extrapolation etc.

Thank you very much, we substantially improved the abstrac following the suggestions

Introduction

This part also comprise of also a review of studies where similar methods of modeling has been implemented for predictions and future guidelines of relevant stockholders. Please follow the below given suggestions for betterment of studies.

1. Line 43-53 you talked only about climate change. Either include in your title the climate change or summarize these lines.

Thanks for the suggestion, we removed the part that includes climate change and we refer only to climate variability to avoid confusion for readers

2.Niño-Southern Oscillation (ENSO) phenomena in the form of a mini review where this phenomenoa already has been used as a parameter in modeling procedure must be the part of introduction.

Thanks for the suggestion, but we consider that we include a brief but pertinent explanation of what the ENSO cycle consists of and why it is important to evaluate it from the ecological niche modeling approach

3.Line 64- 68 you must come to your studied species of insect and must discus in two lines the effect of environmental variables on the biology of the studied species.

Thank you, we include the genus anastrepha in this paragraph to further specify the work

4.Shorten line 75-97, 99-122.Here the theme presented must be shorten then add mini review about modeling species The maxent implementation in the relevant studies in surrounding countries must be the part of these lines too.

Thank you very much, we reduced the paragraph by eliminating irrelevant information about the species, we did not include details of previous work in Latin America due to the fact that there are no specific works on climate variability and ecological niche models using Anastreha as model group

Materials and methods

1.Study area map should be included.

Thanks for the suggestions. The map of the study area was included.

2.Line 155 seems repetition

Thank you, following the suggestion we deleted this part of the paragraph “ENSO is a natural phenomenon that has a great influence on the climatic conditions of various parts of the world”

3.How final maps were prepared from output of maxent is missing. How you got averages see line 205-206.

Thanks for the suggestion, we included a short methodological explanation of how we got the binary maps from the Maxent output.

Results

1.All figures need improvement for readers

Thanks, we improved the figures

2.Niche overlap analysis may be included.

This suggested analysis is very interesting and is part of a larger project, but it was not the objective of this manuscript, here we only work with geographic space.

3.Areas may be calculated in kilometers

Thank you, but we consider that the value of change in area represented in percentage is easier for readers to interpret

Discussion

This part has been written well.

Thanks

Conclusion

This part has been written well.

Thanks

References. All refrence must be checked for journals format.

1.Line 555. 624, 649, seems space issue?????

2.Line 738,754 just observe you are not following one format while writing the journal name. Kindly follow only single format suggested by the journal.

3.Line 668; ROSADO ??????????????????????????.

Thank you, following the suggestion we check and correct the style of all the references

General Comments. The manuscript has been written well. Slight changes are needed for betterment.